# The Challenging Life of Mutators: How *Pseudomonas aeruginosa* Survives between Persistence and Evolution in Cystic Fibrosis Lung

**DOI:** 10.3390/microorganisms12102051

**Published:** 2024-10-11

**Authors:** Martina Rossitto, Valeria Fox, Gianluca Vrenna, Vanessa Tuccio Guarna Assanti, Nour Essa, Maria Stefania Lepanto, Serena Raimondi, Marilena Agosta, Venere Cortazzo, Vanessa Fini, Annarita Granaglia, Enza Montemitro, Renato Cutrera, Carlo Federico Perno, Paola Bernaschi

**Affiliations:** 1Multimodal Laboratory Medicine, Bambino Gesù Children’s Hospital, IRCCS, 00165 Rome, Italy; martina.rossitto@opbg.net (M.R.); valeria.fox@opbg.net (V.F.); 2Microbiology and Diagnostic Immunology Unit, Bambino Gesù Children’s Hospital, IRCCS, 00165 Rome, Italy; vanessa.tuccio@opbg.net (V.T.G.A.); nour.essa@opbg.net (N.E.); mariastefania.lepanto@opbg.net (M.S.L.); serena.raimondi@opbg.net (S.R.); marilena.agosta@opbg.net (M.A.); venere.cortazzo@opbg.net (V.C.); vanessa.fini@opbg.net (V.F.); annarita.granaglia@opbg.net (A.G.); carlofederico.perno@opbg.net (C.F.P.); paola.bernaschi@opbg.net (P.B.); 3Pneumology and Cystic Fibrosis Unit, Bambino Gesù Children’s Hospital, IRCCS, 00165 Rome, Italy; enza.montemitro@opbg.net (E.M.); renato.cutrera@opbg.net (R.C.)

**Keywords:** *Pseudomonas aeruginosa*, cystic fibrosis, hypermutability, whole-genome sequencing, evolution, multi-drug resistance, modulator

## Abstract

Cystic fibrosis (CF) is a life-threatening genetic disease characterised by chronic lung infections sustained by opportunistic pathogens such as *Pseudomonas aeruginosa.* During the chronic long-lasting lung infections, *P. aeruginosa* adapts to the host environment. Hypermutability, mainly due to defects in the DNA repair system, resulting in an increased spontaneous mutation rate, represents a way to boost the rapid adaptation frequently encountered in CF *P. aeruginosa* isolates. We selected 609 isolates from 51 patients with CF chronically colonised by *P. aeruginosa* to study, by full-length genome sequencing, the longitudinal evolution of the bacterium. We recovered at least one hypermutable (mutator) isolate in 57% of patients. By combining genomic information and phenotypic analyses, we followed the evolutionary pathways of the *P. aeruginosa* mutator strains, identifying their contribution to multi-drug resistance and the emergence of new sub-lineages. By implementing patient clinical data, we observed that mutators preferentially follow a specific evolutionary trajectory in patients with a negative clinical outcome and that maintenance antibiotic polytherapy, based on alternating molecules, apparently reduces the occurrence of hypermutability. Finally, we draw attention to the possibility that modulator-induced changes in the pulmonary environment may be associated with the onset of hypermutability.

## 1. Introduction

*Pseudomonas aeruginosa* chronically colonises the lungs of a patient with cystic fibrosis (CF), leading to increased morbidity and mortality. Hypermutable (or mutator) strains of *P. aeruginosa* have often been found in people with CF (pwCF), with prevalence in isolates from pwCF varying across studies, and in continents, around averages of 27–29% [1]. Hypermutability enhances the genetic diversity of the *P. aeruginosa* population in CF lungs through increased accumulation of new mutations, an advantageous feature that allows for rapid adaptation to a variety of stressful environments (e.g., antibiotic and/or immune system pressure, as well as coexistence with co-infecting microorganisms) [1,2]. Increased mutation rates arise from mutations in genes involved in the mismatch repair (MMR) system (m*utS*, *mutL*, and *uvrD*) or in the 7,8-dihydro-8-oxo-deoxyguanosine (8-oxodG or GO) system (*mutM*, *mutY*, and *mutT*) [1], with the former mechanism frequently responsible for increased mutation rates in CF isolates [3]. Mutators can coexist with non-mutator isolates (accumulating on average 2.6 SNPs/yr) in CF lungs, potentially through niche separation [3]; however, mutators have been observed to dominate the entire population colonising some patients with CF [2]. Finally, despite mutator strains being strongly linked with antibiotic resistance development both in vitro and in vivo in patients with CF [2], there is a lack of concordance on their clinical impact across studies [3]. 

Starting from a large phenotypic and full-genomic characterisation study involving hundreds of *P. aeruginosa* strains isolated longitudinally from 51 pwCF in follow-ups at our hospital, we analysed the detected mutator population in terms of prevalence, multiresistance, evolutionary trajectory, and response to particular therapeutic approaches to which our patients were subjected (e.g., modulators). Continuing to add information on these subpopulations of *P. aeruginosa* that are frequently encountered in pwCF will help in shedding light on their clinical impact and enable a better management of pwCF, especially for those who (for now) cannot benefit from modulator therapy. 

## 2. Materials and Methods

We selected 609 *P. aeruginosa* strains collected from 2004 to 2023 from 51 chronically colonised pwCF. Frozen bacterial stocks were plated on Columbia agar + 5% sheep blood (bioMérieux, Marcy l’Etoile, France) and MacConkey agar and incubated overnight at 37 °C to verify the phenotype. Antibiograms were performed by the broth microdilution method using the MicroScan panel NM-NF64 (Beckman Coulter, Indianapolis, IN, USA) and were interpreted according to clinical breakpoints based on the European Committee on Antimicrobial Susceptibility Testing (EUCAST) tables (version 14.0) (EUCAST, n.d.).

Whole-genome sequencing was performed on all 609 *P. aeruginosa* strains to assess sequence types (STs), resistome and virulome. Bacterial DNA was extracted using the automatic extractor EZ1 (Qiagen BioRobot EZ1), with the proper extraction kit (EZ1&2 DNA tissue kit, Qiagen, Hilden, Germany), following the manufacturer’s instructions and setting the elution volume at 50 µL. Extracted DNA was quantified using Bioanalyzer Instrument, and Next Generation Sequencing library preparation was performed according to the manufacturer’s protocol with the DNAprep kit (Illumina, San Diego, CA, USA). Prepared libraries were sequenced with an Illumina NextSeq 550 sequencing platform using a NextSeq 500/550 v2.5 Kit to obtain 2 × 150 bp paired-end reads.

Raw reads were trimmed for adapters and filtered for quality (Phred score > 28) using Fastp (v0.23.4, [4]) and quality checked after trimming with FastQC (v0.11.9, [5]). Kraken2 (v2.1.3, [6]) was used to determine the taxonomic classification and screen for potential contaminations. Whole-genome sequencing reads were assembled de novo using Shovill (v1.1.0, [7]), and the quality of the assemblies was evaluated using Quast (v5.1, ([8]) and was annotated using Prokka (v1.14.6, [9]). Multilocus Sequence Typing (MLST) was performed with MLST tool (v2.11, [10]), and the serotype was inferred with PAst [11]. The investigation of antibiotic resistance (AMR) genes and virulence factors was carried out with ABRicate (v0.4, [12]) by using the Comprehensive Antibiotic Resistance Database (CARD, [13]) with 90% coverage (–mincov) and 90% identity (–minid) parameters, while virulence factors using the Virulence Factor Database (VFDB, [14]) were investigated by using said database with 70% coverage and 70% identity parameters. Mutations in MMR, GO, and antimicrobial resistance genes were described by mapping the reads with BWA mem [15] against the PAO1 reference (GenBank acc n NC_002516.2). Single-nucleotide polymorphism (SNP) calling was performed with Snippy (v4.6.0, [16]), using the default parameters and the PAO1 genome (GenBank accession number NC_002516.2) as reference. To define a strain as a mutator, a threshold of >50 SNPs/yr compared to the ancestral strain was applied [3].

Clinical data were retrieved from patient records. This study received approval from the ethics committee of the Bambino Gesù Children’s Hospital (protocol number 2917_OPBG-2022).

## 3. Results

### 3.1. Prevalence and Resistance Profiles

We recovered at least one mutator isolate in 57% of the 51 selected patients, accounting for 11.5% (70 strains) of the 609 *P. aeruginosa* isolates analysed. Thus, 29 patients had one or two colonising STs in a hypermutable state at least once in their life. Indeed, among these 29 patients, a total of 315 strains were retrieved, with 70 (22%) showing mutator phenotypes. 

Multidrug-resistant (MDR) and extensively drug-resistant (XDR) phenotypes were observed in 57 out of 70 mutator strains (Appendix A). In some patients, nonsynonymous (missense or stop) mutations in *mutT* and *mutL* seemed to be linked to the appearance of MDR in previously non-resistant STs (Appendix A). In particular, in six patients, the colonising clones developed multi-resistance following the appearance of hypermutability, while in twenty patients, the mutator phenotype arose in already multi-resistant clones. Finally, in three patients, the appearance of the mutator was not associated with the emergence of resistance. Overall, the most mutated resistance genes across the different ST were *ampC*, *armZ*, *mexD*, *mexS*, *mexT*, *mexY*, *nalC*, *oprD*, *oprJ*, *PBP3A* (*pbpC*), *pmrA*, *pmrB*, and *rpoB* (Figure 1).

All the 70 strains showed mutations in *mexS* (involved in carbapenem resistance), *mexY* (involved in aminoglycoside resistance), *nalC* (a regulator of the MexAB-OprM system, involved in quinolone and beta-lactam resistance), and *pmrB* (part of the two-component system PmrAB, involved in polymyxin resistance). Phenotypic resistance to colistin was observed only in eight mutator strains; while three strains did not have different *pmrB* allele from colistin-sensitive isolates, five resistant strains showed unique mutation combinations—A248T;Y345H, F168L;Y345H (two strains, same patient), R10L;Y345H, and V185A;Y345H 

Overall, the number of mutations in the resistome genes remained roughly constant during the years (up to 12) of persistence of the mutator strains in the patients (Appendix A). 

### 3.2. Mutations Responsible for the Mutator Phenotype

We observed a direct correlation between SNP increase and the presence of nonsynonymous mutations in MMR or GO systems in 35 mutator isolates (Appendix A). Hypermutability was likely caused by a single mutation/combination of mutations in a gene involved in the MMR or GO systems, namely *mutS* (20 strains, nine STs), *mutL* (11 stains, six STs), and *mutT* (4 strains, three STs), and SNPs appeared to increase differently depending on the gene involved. Excluding mutator isolates with mutations in multiple genes, we recovered an average of 1135 SNPs (range: 813–1573, 3 isolates) in isolates with mutated *mutT*, 435 SNPs (range: 151–1328, 15 isolates) in isolates with mutated *mutS*, and 237 SNPs (range: 62–488, 9 isolates) in isolates with mutated *mutL.* When mutations affected multiple genes simultaneously, we observed some of the highest SNP accumulations; simultaneous mutations in *mutL*, *mutS*, and *urvD* resulted in 2215 SNPs with respect to the ancestor in one isolate, whereas the association of mutations in *ung*, *mutT*, *mutL*, and *mfd* led to 1458 and 1635 SNPs in two isolates (one ST, one patient). Among the remaining isolates, the probable cause of the increase in SNPs was identified in mutations in MMR or GO genes and other mutator-associated genes (*mfd*, *radA*, *sodM*) for an additional nine isolates. 

Four patients shared the ST274, which overall presented an inter-patient SNP/year average of 87 (range 60–129) and an average of SNPs with respect to the ancestor of 401 (62–874). Of the 10 mutator strains, 8 strains showed MDR phenotypes. In all the patients, this clone presented the same allelic variants in *mutM* and *mutT*, whereas it had the same stop mutation Q135* in *mutL* in the three strains (two patients) with the lowest SNPs vs. ancestral (62–118) (Appendix A). Nine patients shared the ST3243, which overall presents an inter-patient SNPs/year average of 89 (range 75–101) and an average of SNPs with respect to the ancestor of 307 (149–571). All the 11 mutator strains showed MDR or XDR phenotypes. An increase in SNPs was observed in association with mutations in *mutS* (two patients), *mutL* (one patient), *urvD* (one patient) and *sodM* gene (one patient) (Appendix A). Thus, with the exception of strains belonging to ST274 with identical mutations suggesting the transmission of the mutator strain in two patients, in the other cases, both STs appear to have independently adopted a mutator phenotype during the colonisation of each of the patients.

### 3.3. Mutators Evolutionary Trajectories

After their appearance, mutators took different evolutionary pathways, ranging from evolution into new sub-lineages (giving rise to the clonal complex within patients) to reversion to the ancestral normo-mutable rates. 

In six patients (four different STs and the ST3243 shared between two patients), hypermutability precedes the appearance of new sub-lineages in the immediately following years (Appendix A). In four patients, two to three new STs were observed (ST4903-4904, ST4947-4948-4949, ST4951-4952, and ST4959-4960), always preceded by SNP increase in the progenitor isolate. In the two patients sharing ST3243, we observed the evolution of a new lineage each (ST4953 and ST4961) and the simultaneous reversion of the ancestral clone to a non-mutator state. In three patients, mutator isolates showed a nonsynonymous mutation in *mutS;* one patient showed a mutation in *radA*, while no clear mutations were identified for the remaining two patients. In a seventh patient, the new ST4950 was retrieved years before isolation of the high-risk epidemic clone ST274 from which it appears to be derived. This new ST is the only one of our cohort showing a stop mutation in *mutL;* however, we were unable to verify which mutation in ST274, if any, caused its appearance. All the new sub-lineages conserved the MDR or XDR phenotype of the evolving clone. Thus, hypermutability seems to be a key driver in the emergence of new sub-lineages within patients, often leading to the rapid diversification of sub-lineages. This evolutionary path contributes to the genetic diversity seen in chronic infections but retains the resistance phenotype, potentially allowing the pathogen to persist, as different sub-lineages may adapt to different niches within the host.

In 20 patients, the mutator phenotype stopped accumulating > 50 SNPs/year after several years (up to 10) without a concomitant loss of the mutation responsible for the onset of hypermutability. In seven patients, this persisting mutator coexisted with a non-mutator ancestral. Overall, in 69% of the patients with mutators, this phenotype appears to fix the overall number of SNPs vs. the ancestor, persisting for years in this form without reverting to the normal mutation rate and without any observable complementation of previous mutations. The persistence of mutators was observed in all of the nine patients with unfavourable outcomes (six dead and three transplanted) (Appendix A). 

Finally, we observed mutator reversal to a non-mutator phenotype, after losing their respective mutations in *mutT*, *mutS*, or *urvD*, in six patients, as well as the disappearance of mutators with the highest SNP/year level (506 and 1505) in two patients (Appendix A).

We did not find a correlation between the choice of these evolutionary pathways and the possible co-presence of other pathogens in the lung. Indeed, we were able to trace back the co-colonisation in the years of the onset of hypermutability of *P. aeruginosa* for 25 patients. Among these patients, 15 exhibited a chronic co-colonisation sustained by moulds and/or yeasts (mainly *Candida albicans*, *Aspergillus fumigatus*, and *Scedosporium apiospermum*), 10 by methicillin-sensitive *Staphylococcus aureus*, and 5 by methicillin-resistant *S. aureus.* Finally, two patients were colonised, one chronically and the other sporadically, by *Mycobacterium abscessus.* For the 14 patients who displayed hypermutability for more than 2 years, co-colonisation was stable and unaffected by evolutionary trajectories undertaken by *P. aeruginosa.*

### 3.4. Modulators and Hypermutability

To assess the possible link between hypermutability and maintenance antibiotic therapy, we followed back the therapeutic approaches to which the 51 patients were subjected as standard of care in the maintenance treatment of chronic *P. aeruginosa* infection. We were able to recover sufficient data for 25 out of 29 patients with mutators and 15 out of 22 patients among those without mutators. Of the first group, 20 had been on antibiotic monotherapy for years, i.e., only azithromycin or only an antibiotic per aerosol without rotation of different molecules (e.g., colistin and tobramycin). Among the 15 patients with no hypermutations, only 5 were on monotherapy and 10 were on polytherapy. 

Finally, we studied the appearance of mutators under therapy with modulators. Mutators appeared under modulators therapies in three patients (four strains). In particular, two patients developed hypermutability under Lumacaftor/Ivacaftor (Orkambi^®^), due to mutation in *mutT* or *mutS* in three strains (2 STs). In one patient, a single mutation in *mutT* appeared to be associated with a 5-fold increase in SNPs in 4 years (from 259 to 1573 SNPs) and the development of multi-resistance, with mutations arising in most of the resistance genes (see PA_02_10 and PA_02_11 of Figure 1). In the second patient, the mutation in *mutS* was lost after 2 years from the non-mucoid strain in which it had arisen, and the normo-mutation rate was again observed; it persisted after switching to Elexacaftor/Tezacaftor/Ivacaftor (ETI). In this particular strain, the loss of mutator status coincided with the loss of the MDR phenotype. In the third patient, one strain developed hypermutability under ETI; for this strain, no mutations were identified to explain the 6-fold increase in SNPs. In a further six patients under ETI, mutator SNPs were reduced to various levels for five patients and remained unchanged in one patient. 

## 4. Discussion

The prevalence of mutator clones in CF chronic *P. aeruginosa* infections is an issue of a potential great impact in infection management, given their link to antibiotic resistance, and perhaps to the evolution of the bacterium-driven damage [17]. As part of a larger study on the longitudinal evolution of *P. aeruginosa* during the years-long colonisation of pwCF, we found that 57% of patients had a mutator phenotype at least once in their lifetime. The prevalence of *P. aeruginosa* mutators among CF patients has been analysed in several studies, with different percentages of patients harbouring mutator isolates [1]. Our observation is similar to Danish CF prevalence (54%) [18] and explainable by the fact that, for many of the patients, we analysed strains already part of an advanced stage of colonisation, while the Danish study also investigated isolates from early stages of infection. 

More than 80% of our mutator strains were MDR or XDR, but they accounted for only 15% of the resistant strains collectively recovered in our 51 patients. Therefore, our data contradict the significant correlation between hypermutation and antibiotic resistance (i.e., mutators are more frequently antibiotic-resistant than non-mutators previously reported [19]. However, among the 397 antibiotic-resistant strains colonising our 51 patients, 251 isolates belonged to epidemic high-risk clones (e.g., ST274, ST395) or others clones already circulating as MDR and not necessarily expressing the mutator’s SNP levels. Furthermore, 127 MDR/XDR strains appeared to be descendants of a mutator strain that retained approximately the same number of SNPs towards the ancestor but reduced the SNP/year accumulation rate (hence no longer considered mutators in our analysis). Although it has been suggested that high mutation rates may increase the likelihood of resistance reversal [20], we did not observe this tendency among our mutators that instead keep the number of mutations in the resistome accumulated over the years stable. On the contrary, we observed a loss of multiresistance associated with reversion to a non-mutator phenotype. Thus, by contributing to maintaining a high level of antibiotic resistance in patients, mutators worsen the already high probability of multiresistance in patients with CF, hindering the success of antibiotic treatment.

Our evidence of low colistin resistance prevalence among mutators (1.3% vs. 6.6% of non-mutators) supports observations that reported no increased resistance among mutator strains to this last-resource option [19,21]. Nevertheless, in line with observations that sub-inhibitory concentrations of colistin can induce hypermutation [22,23], the use of colistin as maintenance monotherapy does not seem to protect against the onset of hypermutability in our cohort. Our observation points towards the use of combination therapies for the maintenance of chronic infections; however, further confirmations are still required. Indeed, it has been observed that in mixed biofilms (as in CF), resistant bacteria offer protection to susceptible ones under the pressure of antibiotics [23], and, after discontinuing monotherapy or switching to alternative antibiotics, a more suitable population may re-emerge.

As previously observed [24], mutations in *mutT* lead to a strong mutator phenotype, defined in our case by the highest SNP average, while *mutS* and *mutL* appeared to be linked with lower levels of SNP accumulation. Interestingly, when *mutS* (but no *mutL* or *mutT*) was involved in the onset of hypermutability, we witnessed the evolution of new sub-lineages. Nonetheless, even when MMR system genes were not clearly involved, each occurrence of a new sub-lineage in our cohort (11 new STs) was preceded by a hypermutability phase of the progenitor ST. As a side note, of the six patients in whom a mutator evolved in a new ST, only one had a negative outcome. This patient was first transplanted and then eventually died; he had two mutator populations of the same ST, with different mutations in multiple MMR genes (and consequently high SNPs) following two different paths. One evolved in a new sub-lineage, while the other persisted with high SNPs. Our data lead us to speculate that the evolution of a new sub-lineage may indicate a newfound equilibrium within the host environment, unless deeply unstable elements persist (with the continuous accumulation of very high SNPs), and this achieved balance could be reflected in a lower clinical impact. Whether the appearance of new lineages marks, or is a manifestation of, a new equilibrium between host and bacterium needs further investigation. If confirmed, this could be taken into consideration in the clinical management of patients, both for those whose mutator evolves to sub-lineages and, particularly for those in whom persistence, rather than evolutionary divergence, is the preferred path.

Mutators are generally expected to return to the non-mutator state to avoid the double-edged sword effect of collateral genomic damage resulting from the accumulation of random mutations [25]; however, reversion is not frequently observed in CF *P. aeruginosa* [26]. Consistently, only 10% of the mutator strains identified in this study returned to their original non-mutator state. This contributes to the hypothesis that hypermutation, that in the short-term allows for rapid exploration of the ‘adaptive landscape’ under a new selective pressure, can have a negative impact on long-term bacterial fitness, when harmful mutations are added to beneficial ones [25]. Indeed, mutators with high SNP levels that did not equilibrate with the host environment (e.g., through the evolution of new STs or fixation of SNP levels) were lost, at least in the patients still alive in our cohort, suggesting they were accumulating detrimental mutations. 

Conversely, mutator fixation may be advantageous because of the accumulated adaptive mutations and the continuous selective pressure exerted by the CF lung environment [26]. Consistent with this observation, we have recorded a tendency for the mutator strain to become fixed in the population once it has presumably reached the necessary degree of adaptation (69% of patients). This stabilisation occurs without reversion to the non-mutator phenotype but by maintaining the levels of SNPs accumulated during the hypermutability period (up to 12 years). Indeed, mutator persistence was observed in 20 patients, 9 of whom had a negative outcome (5 dead, 3 transplanted, and 1 transplanted and eventually dead), while reversion to the non-mutator state was not recorded in any of the patients with a negative outcome. The impact of persistence on the disease trajectory needs to be further investigated. Mutator stabilisation could be the optimal adaptative strategy that allows bacteria to survive in the extremely stressful and fluctuating lung environment, which, when prolonged, leads to lung tissue damage in patients with CF.

In seven cases, the coexistence of mutator and non-mutator strains derived from the same ancestral clone was observed. Moreover, in 13 of the 15 patients colonised by two unrelated clones, only 1 adopted the mutator phenotype. These data suggest that hypermutability is not a strategy that all colonising strains necessarily adopt to cope with the harsh CF environment. The reason for this may lie in the colonisation of different lung niches, characterised by a highly diversified environment and subjected to different environmental stress conditions [26] or, equally plausible in our opinion, by the different mutations exhibited by coexisting clones affecting *lasR* of the quorum-sensing (QS) *las* system. Indeed, QS-mediated regulation of mutators has been recently described for polymicrobial communities [27], and its role on hypermutability appearance in polyclonal *P. aeruginosa* colonisations needs further investigation. Nonetheless, the continuous alteration of CF lung anatomy forces bacterial adaptation also prompting adaptive radiation, not only to different morphotypes but also to new sub-lineages, as we witnessed. Furthermore, we observed that a single clone can simultaneously take two opposite evolutionary paths in the same patient; following a hypermutability phase, a new sub-lineage appeared, while a reversion of the ancestral clone towards a non-mutator phenotype was observed. Interestingly, this occurred in only two patients, both carrying ST3243, and we can speculate that spatial separation and colonisation of different niches, combined with typical features of this clone, may be the cause of this divergence. 

Our analysis revealed no significant correlation between the evolutionary pathways of *P. aeruginosa* and the possible co-presence of other pathogens in the lung environment. This finding aligns with previous studies, which experimentally demonstrated that *P. aeruginosa* hypermutability has little impact on interspecies competition in the short term [25]. Our preliminary data seem to corroborate these findings and to extend this observation also in in vivo settings and over long-term co-colonisation periods (up to 10 years). Nonetheless, further investigation on the mechanisms underlying these interactions is needed, also to better understand polymicrobial infections and aid in the definition of potential therapeutic strategies targeting these co-colonising pathogens.

Finally, our observation that 50% of patients taking Orkambi^®^ developed mutators and, in one patient, multi-resistance, highlights the need for careful evaluation of the effect that modulators may have on patient-colonising clones. In particular, chronically colonised pwCF treated with modulators should not be considered microbiologically ‘cured’, given the emergence of hypermutable and resistant phenotypes that may appear when bacteria adapt to the changing CF lung environment. Nonetheless, it is true that this increase in SNPs was observed only in one patient (out of twenty-five) on ETI, and this could reflect the greater clinical efficacy achieved by this recent modulator compared to earlier ones [28]. However, it should be considered that bacterial adaptation through hypermutability may not be a direct consequence of the modulator itself but rather of the simplification of other maintenance treatments (with or without medical consent), resulting in a different selective pressure. The long-term effect of modulator therapies on airway infections is currently being studied by the scientific community [29,30], and the impact on chronic *P. aeruginosa* colonisation needs to be thoroughly evaluated in the future, with a focus on the possible effects on mutators and consequent disease progression and outcome. 

## 5. Conclusions

Mutators in CF are generally assumed to contribute to antimicrobial resistance increase, although recent studies have also shown important effects on the evolution of virulence, genetic adaptation to the CF lung environment, chronicity, and possibly decline in lung function [24]. Our results, while requiring confirmation on larger numbers, suggest that reversion to a normo-mutable state or evolution into a new ST may act as protective factors towards the host, unless mutators with markedly elevated SNP levels are co-infecting. Of course, we cannot exclude that changes in the host condition, independent of the bacterium’s state, determine the evolutionary fate of the mutator. Further studies are needed to clarify this point of major clinical relevance. 

This present work, while adding information to the vast landscape of mutators in CF, highlights the need for further studies that thoroughly evaluate the impact of mutators on disease progression and outcome, also considering the importance of properly using new therapeutic approaches, both in terms of the duration of treatment and of the number of pwCF treated with these modulators. 

## Figures and Tables

**Figure 1 microorganisms-12-02051-f001:**
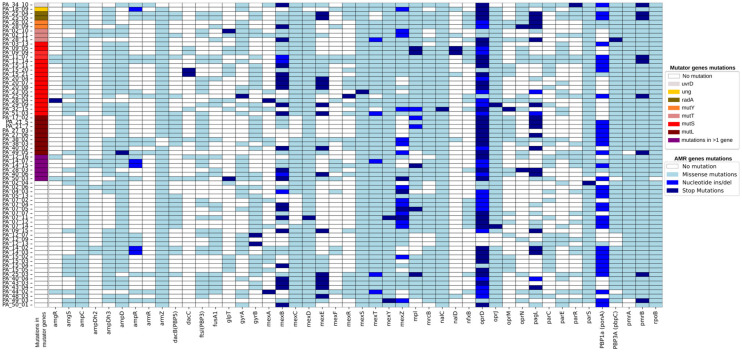
The mutations in the resistome (resistance-associated genes) identified via whole-genome sequencing and bioinformatics analysis for the 70 mutator *Pseudomonas aeruginosa*. Mutator genes involved and typology of mutations (missense, nucleotides insertions/deletions, or stop) are shown in different colours and are reported in the legend.

## Data Availability

The original contributions presented in the study are included in the article/Appendix A, further inquiries can be directed to the corresponding author.

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
