# Peer review of "The Challenging Life of Mutators: How Pseudomonas aeruginosa Survives between Persistence and Evolution in Cystic Fibrosis Lung"

_microorganisms, 2024, doi:10.3390/microorganisms12102051_

Round 1

Reviewer 1 Report

Comments and Suggestions for Authors

Dear authors, I had the pleasure of reading your manuscript entitled " The challenging life of mutators: how Pseudomonas aeruginosa survives between persistence and evolution in cystic fibrosis lung ".  After reading your manuscript I have some comments.

 1.    Paragraphs 105-109 present some inconsistencies regarding the number of mutators. Initially, it is stated that 11% of the 609 Pseudomonas isolates are mutators (67 in total). Subsequently, it is mentioned that of 315 isolates, 22% exhibit a mutator phenotype (70 in total). This discrepancy raises the question of the actual number of mutators. Furthermore, clarification is needed on the phenotypic definition of a mutator.

2.    It would be beneficial to include a table presenting the resistance profiles of the strains in Figure 1 or Table 1.

 I hope that these comments will be helpful in improving the manuscript.

Author Response

Comments 1: Paragraphs 105-109 present some inconsistencies regarding the number of mutators. Initially, it is stated that 11% of the 609 Pseudomonas isolates are mutators (67 in total). Subsequently, it is mentioned that of 315 isolates, 22% exhibit a mutator phenotype (70 in total). This discrepancy raises the question of the actual number of mutators. Furthermore, clarification is needed on the phenotypic definition of a mutator.

Response 1: We thank the reviewer for pointing this out. We realized that the misunderstanding comes from rounding the percentage, since 70 is in fact 11.49% of 609. To avoid misunderstanding we have changed it to 11.5% and reported the actual number in brackets. We also modified the paragraph to clarify it.

We reported the definition of mutator in lines 98-99.

Comments 2: It would be beneficial to include a table presenting the resistance profiles of the strains in Figure 1 or Table 1.

Response 2: We thank the reviewer for the suggestion. We have thus included a new table as supplementary material showing the resistance profile of the 70 strains.

Reviewer 2 Report

Comments and Suggestions for Authors

Pseudomonas aeruginosa infection often accompanies chronic progressive lung disease, a major cause of mortality in cystic fibrosis (CF) patients. P. aeruginosa isolates adapt to the CF lung environment during infection through a series of profound genotypic and phenotypic changes to ensure bacterial survival by generating maximum diversity. One such mechanism that may provide advantages for colonization of anatomical niches is hypermutation. The authors studied the genomes of 609 P. aeruginosa strains. These strains were collected from 51 chronically colonized patients with cystic fibrosis. They analyzed the identified mutant population with respect to frequency, multiresistance, evolution and response to specific therapies. The research methods are chosen correctly and are described in sufficient detail. The results are discussed in detail. The patterns found in the study are hypothetically explained by the authors.

Minor concern. If the authors had included an analysis of the pressure of co-infecting microorganisms, the discussion would have been enriched. For example, Yue Yuan On and co-authors (On, Y.Y., Figueroa, W., Fan, C. et al. Impact of transient acquired hypermutability on the inter- and intra-species competitiveness of Pseudomonas aeruginosa. ISME J 17, 1931–1939 (2023). https://doi.org/10.1038/s41396-023-01503-z) showed that acquired hypermutability primarily provides a competitive fitness advantage within species.  

Author Response

Comments 1: If the authors had included an analysis of the pressure of co-infecting microorganisms, the discussion would have been enriched. For example, Yue Yuan On and co-authors (On, Y.Y., Figueroa, W., Fan, C. et al. Impact of transient acquired hypermutability on the inter- and intra-species competitiveness of Pseudomonas aeruginosa. ISME J 17, 1931–1939 (2023). https://doi.org/10.1038/s41396-023-01503-z) showed that acquired hypermutability primarily provides a competitive fitness advantage within species. 

Response 1: We thank the reviewer for the valuable suggestion.  Agreeing that it adds value to our work, we have expanded Results section (lines 202-211)and the Discussion (lines 341-350) to address this aspect.